Validity and usefulness of the student-athletes’ motivation toward sport and academics questionnaire: a Bayesian multilevel approach

Quinaud Ricardo T. 1
Gonçalves Carlos E. 2
Possamai Kauana 1
Morais Cristiano Zarbato 1
Capranica Laura 3
Carvalho Humberto M. humberto.m.carvalho@ufsc.br hmoreiracarvalho@gmail.com 1
1 Department of Physical Education/ School of Sports, Federal University of Santa Catarina , Florianópolis , Santa Catarina , Brazil
2 Faculty of Sport Sciences and Physical Education, University of Coimbra , Coimbra , Portugal
3 Department of Movement, Human and Health Sciences, University of Rome “Foro Italico” , Rome , Italy
Shang Yilun
Electronic publication date: 2021 Jul 30
Publication date: 2021
Volume: 9
Electronic Location ID: e11863
Received 2021 Mar 25; Accepted 2021 Jul 5
Copyright: ©2021 Quinaud et al.
Copyright year: 2021
Copyright holder: Quinaud et al.
License: This is an open access article distributed under the terms of the Creative Commons Attribution License, which permits unrestricted use, distribution, reproduction and adaptation in any medium and for any purpose provided that it is properly attributed. For attribution, the original author(s), title, publication source (PeerJ) and either DOI or URL of the article must be cited.
License URL: https://creativecommons.org/licenses/by/4.0/

Keywords: Dual-career, Sports, Education, Behavior, Bayesian multilevel regression, Post-stratification, Higher education

Funding: The Coordenação de Aperfeiçoamento de Pessoal de Nível Superior - CAPES (finance code 001) Ricardo T. Quinaud was supported by a grant from the Coordenação de Aperfeiçoamento de Pessoal de Nível Superior - CAPES (finance code 001). The funders had no role in study design, data collection and analysis, decision to publish, or preparation of the manuscript.

==============================
Background

Reliable assessment and understanding of student-athletes’ motivation for dual careers are crucial to support their career development and transitions. The purpose of this research was to examine the validity and usefulness of the student-athletes’ motivation toward sport and academics questionnaire (SAMSAQ-PT) in the Brazilian higher education context. Four studies were performed.

Methods

In study one, conceptually and semantic translation of the questionnaire and Bayesian exploratory factor analysis were conducted. In study two, a Bayesian confirmatory factor analysis with an independent sample was performed. In study three, Bayesian multilevel modeling was applied to examine the construct validity of the questionnaire in a cross-sectional sample. In study four, the SAMSAQ-PT sensitiveness was examined in a longitudinal sample, and the results were interpreted based on multilevel regression and poststratification.

Results

Altogether the results provided evidence validity and usefulness of the SAMSAQ-PT in Brazilian student-athletes. The Brazilian student-athletes’ motivation scores were sensitive to the influence of sex, sport level, and type of university on career and sport motivation. SAMSAQ-PT estimate scores across an academic year showed a trend of stability in the scores, adjusting for sex, sport level, type of university, and student-athlete status.

Conclusion

The SAMSAQ-PT proved to be a robust and valuable questionnaire, which could be used in Portuguese-speaking countries. The findings of the cross-sectional and longitudinal surveys urge to consider individual and contextual characteristics when investigating motivation of dual-career of athletes, also concerning the sex-related opportunities in university sports. Furthermore, there is a need for a call for action to promote and nurture the student-athletes motivation to remain engaged in both sports and educational commitments.

Introduction

At the university level, student-athletes present several social, cultural, and individual challenges to pursue their sport and education paths (i.e., dual-career), especially at the start of the college degree (Aquilina, 2013; Condello et al., 2019; Gaston-Gayles & Baker, 2015; Ryba, Ronkainen & Selänne, 2015; Simons, Van Rheenen & Covington, 1999). The attention in dual-career, defined as “a career with major focus on sport and study or work” (Stambulova & Wylleman, 2015), has increased in the past years (Stambulova & Wylleman, 2019). Considering the different sports and educational contexts and the various dual-career approaches in place in the European Member States, the European dual-career recommendations urge strategies to foster the student-athletes motivation to pursue their academic and sports achievements (European Commission, 2012). Indeed, motivation is determinant to keep people involved in what they do (Ferdinand & Czernochowski, 2018; Ryan, Bradshaw & Deci, 2019). Thus, understanding student-athletes motivation for dual-career is crucial to support their career development and transitions (Stambulova, Ryba & Henriksen, 2020). In general, dual-career pathways depend on student-athletes’ motivation, identity, health, lifestyle, and wellbeing (Aunola et al., 2018; Breslin et al., 2019; Cartigny et al., 2019; Harrison et al., 2020; Lupo et al., 2017; Martin, 2005; Ryba et al., 2016; Ryba et al., 2017; Sorkkila et al., 2018).

In studying the dual-career motivation of student-athletes from different cultures, the robustness of the psychometric instrument is crucial for cross-cultural comparisons and applied sport psychology (Joshanloo et al., 2014; Sullivan, Murphy & Blacker, 2020; Wu, Lai & Chan, 2014). Since the development of the Student-Athletes’ Motivation toward Sports and Academics Questionnaire (SAMSAQ) in the United States (Gaston-Gayles, 2005) and its validation tested in European (Guidotti & Capranica, 2013; Lupo et al., 2012; Lupo et al., 2015) and Asian (Park, Hong & Lee, 2015) contexts, the effects of different cultures and dual-career support policies have been hypothesized (Ferdinand & Czernochowski, 2018; Fortes et al., 2010; Gaston-Gayles & Baker, 2015; Guidotti & Capranica, 2013; Guidotti, Cortis & Capranica, 2015; Lupo et al., 2012; Lupo et al., 2015; Park, Hong & Lee, 2015). However, a lack of knowledge for Latin America’s countries is still present.

Brazil is the largest country in Latin America. In having a federal structure, Brazil presents contrasting demographic characteristics and cultural backgrounds (Hofstede et al., 2010). While the Brazilian regulation of sports at the federal level, including university sports, was established in 1941 (Brasil, 1941), rules and criteria for the allocation of public resources to the sports sector were established in 1998, assigning responsibility to sports organizations concerning the educational system and vice versa (Brasil, 1998). In particular, the Brazilian University Sports Confederation is responsible for organizing and developing university sports, whereas the Ministry of Education has the primary responsibility of the sports policies allowing athletes to combine their dual-career. Brazilian public higher education institutions enroll around two million students per year (Instituto Nacional De Estudos e Pesquisas Educacionais Anísio Teixeira, 2018). As for private higher education institutions, around six million students enroll each year (Instituto Nacional De Estudos e Pesquisas Educacionais Anísio Teixeira, 2018). Public universities often provide sports infrastructure for students and private universities offer financial support for athletes. However, federal regulation of dual-career policy for student-athletes in Brazil is still not warranted (Carvalho & Hass, 2015). In the absence of a clear dual-career policy in the Brazilian sports system, some sports areas tend to be privileged, and others may be left unattended, leaving gaps in public service coverage. Especially at the state and local levels, programs and actions appear to vary according to different political approaches (Houlihan, 2005). Therefore, differences among Brazilian states and the country’s federalism structure could provide a different level of dual-career support and influence student-athletes’ motivation (Guidotti, Cortis & Capranica, 2015).

There have been problems to replicate psychological results, also referred to as the crisis of confidence (Open Science Collaboration, 2015). With few exceptions (Schweizer & Furley, 2016), sports psychology has overlooked this debate. One of the general debate consequences has been the increased awareness of the limitations and inappropriateness of testing null-hypotheses, establishing statistical significance and p-value use (Amrhein & Greenland, 2018; McShane et al., 2019). Indeed, psychology research deals with complex interactions, noisy measurements, often expected between-individuals heterogeneity, and non-representative and imbalanced samples. To account for different sources of inferential uncertainty, Bayesian methods allow combining the information known before seeing the data (i.e., the prior uncertainty concerning a parameter or hypothesis expressed as a probability distribution) with what is learned from the observed data (i.e., the likelihood of the data conditioned on the parameter or hypothesis) to update knowledge expressed as the posterior distribution (Kennedy & Gelman, 2020; Lee & Wagenmakers, 2013).

Furthermore, the analysis and interpretation in sports psychology research often deal with traditional single-level approaches, albeit with the limitations noted in several scientific areas (Gelman & Shalizi, 2013). A multilevel modeling framework provides a flexible alternative that intuitively considers the data structure and the different sources of variation, providing trustable estimations and predictions for a target population (Gelman & Hill, 2007). The framework has been noted as valuable to advancing cross-culture studies in psychology (Van Hoorn, 2015). Another main advantage of multilevel modeling lies in the natural fit of repeated measures (Singer & Willett, 2003). Considering the lack of longitudinal studies on student-athlete motivation (Stambulova & Wylleman, 2019), multilevel modeling is highly recommended to improve our understanding of Brazilian student-athletes’ motivation toward dual-career.

The present research made use of a Bayesian approach and conducted four studies to examine the validity of the Portuguese version of the harmonized Italian Student-athletes’ Motivation toward Sports and Academics Questionnaire (SAMSAQ-IT/A; Guidotti & Capranica, 2013) and to assess its usefulness in discriminating the influence of academic and sport contexts as sources of variation in the scores in cross-sectional and longitudinal research approaches. In study one, we aimed to translate and explore the psychometric structure of the Portuguese version of the SAMSAQ-IT/A (SAMSAQ-PT) using Bayesian exploratory factor analysis. In study two, we tested the factor structure of the questionnaire that emerged from study one by applying a Bayesian confirmatory factor analysis with an independent sample. Based on the evidence of construct validity, in study three, we aimed to examine the construct validity of the questionnaire in a cross-sectional sample considering sex, sport level, the student-athlete status, and the type of university attended applying Bayesian multilevel regression. Lastly, in study four, we aimed to explore the student-athletes’ motivation scores’ sensitiveness. Hence, we considered a longitudinal measure design across an academic year to analyze changes in motivation scores adjusted for sex, sport level, student-athlete status, and type of university.

Research Design

The research ethics committee of the Federal University of Santa Catarina approved the present research (no. 2.949.805) and voluntary student-athlete provided written consent to participate in the study. The inclusion criteria for recruiting participants encompassed: (1) to be enrolled in a higher education degree; and (2) to compete in organized sports of the Brazilian University Sports Confederation. Data were collected during the Santa Catarina University Games in July 2018 and 2019 (i.e., state Games) and the Brazilian University Games in November 2018 and October 2019 (i.e., national Games). The state games had about 800 athletes, while the national games had about 2.000 athletes participating. Studies one, two, and three comprised participants only from cross-sectional observations. Study four comprised participants from repeated observations across an academic year (measured in 2018 and 2019). Supplementary materials including data and code are available at https://osf.io/cpwdv/.

Study One

This study aimed to translate and explore the psychometric structure of the SAMSAQ-PT using Bayesian exploratory factor analysis. Independent forward and backward Portuguese translations of the SAMSAQ-IT/A (Guidotti & Capranica, 2013) and cognitive interviews were performed to ensure a conceptually and semantic trustable instrument applied in the Brazilian context (Herdman, Fox-Rushby & Badia, 1997; Su & Parham, 2002). Then, the SAMSAQ-PT was administered to a subsample of 74 Brazilian university student-athletes who individually completed the 30-item SAMSAQ-PT, indicating their level of agreement with the statements on a 6-point Likert scale, ranging from 1 (strongly disagree) to 6 (strongly agree). Interviews aiming to ascertain the reasons behind responses were performed to verify the instructions, items, and response options. Therefore, the SAMSAQ-PT was considered suitable to be administered to Brazilian student-athletes.

Methods

Participants

The total sample comprises 862 student-athletes from all five regions of Brazil; however, about 72.6% were from the same region (south). The total sample was split into two independent samples generated through random numbers. In study one, we used the first 50% of the dataset. The sample comprised 248 female and 183 male student-athletes (Mage = 21.7, SD = 3.4 yrs) enrolled in public (n = 200) and private (n = 231) Brazilian universities and competing at international (n = 33), national, (n = 131), state (n = 74) and university (n = 193) levels. Descriptive analysis is presented in Table S1 (available at https://osf.io/cpwdv/).

Data analysis

A Bayesian exploratory factor analysis (BEFA) was conducted, with an initial four latent factors maximum (Kmax) constraint (Guidotti & Capranica, 2013). Then, different factor-structures testing different models were allowed if the original model factor-structure was not confirmed. A total of 60,000 iterations with a burn-in period of 5,000 iterations were run. Default identification restriction (Nid = 1) was used, which lies on the minimum number of manifest variables dedicated to each factor Metropolis–Hastings’ acceptance rate was used to retain items’ posterior probabilities of being different from zero. Considering it is an exploratory analysis, a minimum posterior mean of 3 (cut-off point) was set as an acceptable value to retain an item (Peeters, 2012). Although there is no established recommendation of the minimum acceptable value to retain an item in BEFA, we set it as 3 for convenience (representing a closer interpretation of the frequentist factor loading “0.3”). Thus, items with factor values lower than three were excluded. The BEFA estimates were obtained using the “BayesFM” package (Conti et al., 2014), available as a package in the R statistical language (R Core Team, 2018).

Results

The SAMSAQ-PT presented a three-factor structure (Table 1). Due to low factor loadings (< 3), items 11 (2.74), 18 (1.92), 25 (2.60), and 30 (1.92) were excluded. In particular, seven items loaded the factor named “Sport Motivation” (SM), sixteen items loaded the factor named “Academic Motivation” (AM), and three items loaded the factor named “Career Motivation” (CM).

Table 1 Bayesian exploratory factor analysis (posterior means) of the Portuguese adapted version of the Student-Athletes’ Motivation Toward Sports and Academics Questionnaire.

Item	Factors	
	SM	AM	CM	
1. I am confident that I can achieve a high-grade point average this year (3.0 or above)	5.02		
2. Achieving a high level of performance in my sport is an important goal for me this year	4.61			
3. It is important to me to learn what is taught in my courses		5.51		
4. I am willing to put in the time to earn excellent grades in my courses		5.19		
5. Within an academic environment, I find it more challenging to face difficult tasks		4.42		
6. For me, studies are important to achieve knowledge and skills		5.52		
7. I will be able to use what is taught in my courses in different aspects of my life outside of school		5.38		
8. I chose to play my sport, because it is something I am interested in as a career			3.78	
9. For me, it is important to train seriously to improve my performance	4.77			
10. I chose (or will choose) my major, because it is something I am interested in as a career		4.93		
11. Earning a high-grade point average (27/30 or above) is not an important goal for me this year		2.70*		
12. It is important to me to learn the skills and strategies taught by my coaches		5.24		
13. It is important for me to do better than other athletes in my sport	3.99			
14. The time I spend engaged in my sport is enjoyable to me		4.23		
15. It is worth the effort to be an exceptional athlete in my sport	4.61			
16. The achievement of a degree is important to enrich my knowledge		5.63		
17. In sport, I find stimulating those situations requiring high performances and being difficult to perform		4.87		
18. During the years, I compete in my sport, completing a college degree is not a goal for me	2.09*			
19. I am confident that I can be a star performer on my team this year	4.12			
20. My goal is to make it to the professional level or the Olympics in my sport			3.49	
21. Situations that allow me to test my capacities stimulate me	4.94			
22. I am confident that I can make it to an elite level in my sport (Professional/Olympics)			3.77	
23. I am confident that I can earn a college degree		5.70		
24. I will be able to use the skills I learn in my sport in other areas of my life outside of sports		5.40		
25. Achieving high performances in my sport is not an important goal for me this year		2.44*		
26. For me, it is important to achieve high performances and not to make mistakes	4.62			
27. I am willing to put in the time to be outstanding in my sport	4.25			
28. The content of most of my courses is interesting to me		4.80		
29. It is important for me to obtain a degree, because it will help me to find a job		5.48		
30. It is not worth the effort to earn excellent grades in my courses		1.78*		
Notes.

SM Sport Motivation

AM Academic Motivation

CM Career Motivation

* Items excluded.

Discussion

Considering the translation and psychometric structure of the SAMSAQ-PT, we observed a three-factor structure. This structure is in line with the original American and the versions validated for Emirati, European, and Korean student-athletes (Gaston-Gayles, 2005; Fortes et al., 2010; Lupo et al., 2015; Park, Hong & Lee, 2015). In particular, SM items express the desire for sports performance and for continuing sports career (Keshtidar & Behzadnia, 2017), which is influenced by cultural and educational context for pursuing a sports performance (Fernandes, Moreira & Gonçalves, 2019; Lupo et al., 2015); the AM items mirror the academic commitment of student-athletes, measuring the Brazilian student-athletes’ engagement in an educational/ vocational path concerning a sport or a dual-career path (Cartigny et al., 2019), probably considering the academic degree relevant for future jobs and professional careers (Fortes et al., 2010); and the CM items present a desire for developing a professional sports career (Gaston-Gayles, 2005). Thus, it could be speculated that the Brazilian student-athletes may perceive sport and academic commitments separately.

Study Two

The purpose of study two was to test the 26-item of the 3-factor structure questionnaire emerging from study one by applying a Bayesian confirmatory factor analysis with an independent sample. Specifically, it was intended to provide evidence to allow an informed generalization of the questionnaire factor structure.

Methods

Participants

The participants considered in this study were the other half of the total cross-sectional sample. In particular, 242 female and 99 male student-athletes (Mage = 21.6, SD = 3.5 yrs) enrolled in public (n = 201) and private (n = 230) Brazilian universities and competing at international (n = 26), national, (n = 139), state (n = 86) and university (n = 180) levels.

Data analysis

A Bayesian Confirmatory Factor Analysis (BCFA) was applied to examine the model factorial structure. Two chains for 10,000 iterations with 2,000 used as warm-up, using probabilistic programming language Stan (Carpenter et al., 2017). The model estimations were regularized using normal prior (0, 10) for the manifest variable (intercept) and normal prior (0, 1) for the latent variable were set. A posterior latent variable closer to 0.5 (Merkle & Rosseel, 2018) was set as a satisfactory value to retain an item based on the literature. Moreover, Bayesian root mean square error of approximation (BRMSEA), Bayesian Gamma Hat (BGammaHat), Adjusted Bayesian Gamma Hat (adjBgammahat), and Bayesian McDonald’s centrality index (BMc) were also applied to confirm the model fit (Montenegro-Montenegro, 2020). BRMSEA value close to 0.5 and BGammaHat, adj BGammaHat and BMc values close to one indicate a better fit. The BCFA was conducted using the “blavaan” package (Merkle & Rosseel, 2018) in the R software (RCoreTeam, 2018).

Results

In the BCFA, items 1 (0.28), 10 (0.45), 12 (0.47), 14 (0.25) and 21 (0.38) presented posterior values lower than 0.5 and were excluded. Additionally, this first model did not present good evidence of model fit (BRMSEA = 0.08; BGammaHat = 0.87; adjBgammahat = 0.84; BMc = 0.43). The new model (21 items) was further tested and four items [items 5 (0.30), 17 (0.35), 24 (0.33), and 29 (0.39)] with low factor load values were excluded. In this model, fit indexes were superior, but not well adequate (BRMSEA = 0.07; BGammaHat = 0.93; adjBgammahat = 0.89; BMc = 0.67). Thus, the resulting 17-item model presented factor loadings close or above 0.5 for all items (Table S2, available at https://osf.io/cpwdv/). Only item 13 presented factor loading (0.49) bellow 0.5. Thus, the item was retained. Additionally, fit indexes showed satisfactory adequacy (BRMSEA = 0.06; BGammaHat = 0.96; adjBgammahat = 0.93; BMc = 0.84).

Discussion

In this study, we tested the factor structure of SAMSAQ-PT, which emerged from study one by applying BCFA. Our observations confirmed the three-factor structure but indicated a better structure composed of 17 items for SM (7 items), AM (7 items), and CM (3 items). The confirmatory factor analysis of the SAMSAQ-PT substantiated the three-factor structure (Fortes et al., 2010; Gaston-Gayles, 2005; Guidotti et al., 2013; Lupo et al., 2015; Park, Hong & Lee, 2015), even though to avoid misinterpretations, the Bayesian inference based on strong similarities determined the substantial reduction of the item’s distribution. Different factor structures can be expected due to dual-career policies and social-cultural contexts, as well as to the sport and educational levels of student-athletes (Fortes et al., 2010; Gaston-Gayles, 2005; Guidotti et al., 2013; Lupo et al., 2015; Lupo et al., 2017; Park, Hong & Lee, 2015; Stambulova & Alfermann, 2009). Given the Brazilian miscegenation and the potential influence of European and other cultures on Brazilian student-athletes, both similarities and differences with other cultures were expected.

Regarding the similarities, the AM dimension presented stable comparing to other SAMSAQ validations (Gaston-Gayles, 2005; Guidotti et al., 2013; Lupo et al., 2015; Lupo et al., 2017; Park, Hong & Lee, 2015), although in the present version we had a higher reduction of items. The results may indicate that the academic motivation of higher education student-athletes is a drive based on similar purposes, such as learning (item 3), get good grades (item 4), and the course content (item 28). There were divergences, such as item 16 (the achievement of a degree is important to enrich my knowledge) had loaded in a different dimension than the academic motivation (Lupo et al., 2017). However, this item has a clear academic motivation.

Our results were consistent with the other validations in the SM dimension, but there is less agreement about the items from this dimension across the validation studies (Lupo et al., 2015; Lupo et al., 2017; Gaston-Gayles, 2005; Park, Hong & Lee, 2015). The feeling of being better than other athletes (item 13) is an example of how it can vary between the validations. It represents a sport and career motivation (Lupo et al., 2017), sport motivation (federation structure; Lupo et al., 2015; Gaston-Gayles, 2005), or even academic motivation (Park, Hong & Lee, 2015). In summary, there was a shift of items in SM and CM dimensions in the factor structure of the SAMSAQ-PT. The items with explicit statements focused on professional or athletic careers loaded in the CM. The results suggest that Brazilian student-athletes interpret their motivation for a high level of performance or athletic career as a priority.

Study Three

In this study, we aimed to examine the construct validity of the questionnaire in a cross-sectional sample considering sex, sport level, student-athlete status, and the type of university. In particular, the student-athlete status was considered as any document from the higher education institution that guarantees support for student-athletes (e.g., flexible exam schedule when representing their university or their country in competitions).

Participants

The whole sample of 862 student-athletes (females: 56.8%; males: 43.4%) enrolled in public (n = 401) and private (n = 461) Brazilian universities and competing at international (n = 59), national (n = 270), state (n = 160) and university (n = 373) levels participated in this study.

Data analysis

Multilevel regression models were used to estimate SAMSAQ-PT and its dimensions among Brazilian student-athletes when grouped by sex (e.g., female and male), sports level (e.g., international, national, state and university), type of university (e.g., public and private) and student-athletes’ status (e.g., yes and no). The multilevel models estimate the individual scores relying on the available information of individuals characteristics and using additional “random” predictors such as group or context characteristics (for individual i, with indexes, s, a, l, and u for, sex, student-athlete status, sport level, and type of university, respectively). In Bayesian terms, these “random” or “group-level” effects are related to each other by their grouping structure, and the individuals’ responses are partially pooled towards the group mean (Gelman & Hill, 2007), as follows: yi=β0+αsisex+αaistudent−athletestatus+αlisportlevel+αuitypeofuniversity+ϵi

αsisex∼N0,σsex2, for s = 1, 2.

αaistudent−athletestatus∼N0,σstudent−athletestatus2, for a = 1, 2.

αlisportlevel∼N0,σsportlevel2, for l = 1, 2, 3, 4.

αuitypeofuniversity∼N0,σtypeofuniversity2), for u = 1, 2.

ϵi∼N0,σyi2)

Weakly informative prior distributions, normal prior (0, 10) for population-level effect (intercept), and normal priors (0,1) for group-level effects (i.e., the standard deviations of varying intercepts) to regularize the multilevel model estimations were used. Two chains for 4,000 iterations with a warm-up length of 1,000 iterations to ensure convergence of the Markov chains were run. To check the models and estimations, trace plots to examine the convergence of Markov chains and posterior predictive checks to validate the models were used (Gelman et al., 2013). The Bayesian multilevel models were fitted with the “brms” package (Bürkner, 2017), available as a package in the R statistical language (RCoreTeam, 2018). The brms package implements Bayesian multilevel models using the probabilistic language Stan (Carpenter et al., 2017). For computational and interpretation convenience, the outcomes were standardized (z-scores). This methodology is also described in previous studies (Quinaud et al., 2020; Quinaud et al., 2020b).

Results

Figures 1–3 present the standardized values of the SAMSAQ-PT’s dimensions related to the respondents’ sex, type of universities, sport levels, and student-athletes’ status. Male student-athletes presented higher effect scores for SM and CM dimensions. On the other hand, female student-athletes presented higher scores for AM, albeit with a small magnitude. Student-athletes from private universities had substantially higher scores than student-athletes from public universities for SM, AM, and CM dimensions, but at best, the magnitude was small for AM. Considering sport level variation, student-athletes who competed at higher level of performance showed higher scores than those of lower competitive levels for SM and CM dimensions. Lastly, there was no substantial variation in the motivation scores by student-athlete status. Table S3 (available at https://osf.io/cpwdv/) presents the SAMSAQ-PT estimates and uncertainty (90% confidence intervals).

Figure 1 Posterior values for Sport Motivation dimension by sex, type of universities, competitive levels and student-athletes’ status (67% and 90% credible intervals).

Figure 2 Posterior values for Academic Motivation dimension by sex, type of universities, competitive levels and student-athletes’ status (67% and 90% credible intervals).

Figure 3 Posterior values for Career Motivation dimension by sex, type of universities, competitive levels and student-athletes’ status (67% and 90% credible intervals).

Discussion

We tested the SAMSAQ-PT construct validity by exploring the variation in the motivation scores considering student-athletes’ individual, sport, and academic characteristics. There was substantial variation in the SAMSAQ-PT dimension scores by sex, university type, and sport level. The observed substantial gender-related variation in SM and CM could be due to lower dual career support for Brazilian female athletes and limited economic opportunities for top-level women’s sport (Harrison et al., 2020). Coherently, during the academic path, the highest effect scores for AM emerged for female student-athletes, who might emphasize more their academic career concerning their sports career (Tekavc, Wylleman & Cecić-Erpič, 2015). These findings urge implementing sex-related dual careers at the sport and public education level, as envisaged by the European-funded Collaborative Partnership “Dual Career for Women Athletes” (DONA). Consistent with previous studies (Guidotti et al., 2013; Lupo et al., 2017), the sport level resulted as a predictor of CM and SM scores. It seems reasonable to assume that student-athletes competing at a higher level might be more motivated to pursue their sports career compared to their pairs competing at lower levels, whereas it is plausible to assume that AM is linked to the academic motivation of the athlete (Gaston-Gayles, 2005; Guidotti et al., 2013; Lupo et al., 2015; Lupo et al., 2017).

To our knowledge, this is the first study investigating the student-athletes’ motivation related to university type (e.g., public and private). Indeed, the social context of universities shapes the opinions of student-athletes (Druckman et al., 2014). The present findings highlight that Brazilian student-athletes’ sports and education commitments vary between public and private universities. Concerning their peers from public universities, in the present models, student-athletes from private universities have a high probability of scoring higher values for CM and SM, probably due to dual-career financial and logistic support (Aquilina, 2013; Aquilina & Henry, 2010). Enrolling 75% of the total undergraduate students (Instituto Nacional de Estudos e Pesquisas Educacionais Anísio Teixeira, 2018), private Brazilian universities might employ marketing strategies incorporating student-athletes to increase the media attention leveraging the universities’ profile (Harrison et al., 2010; Teixeira, 2010). Unlike private universities, Brazilian public universities do not require tuition fees and offer academic undergraduate and graduate degrees (e.g., master’s and Ph.D.), opportunities for research, and internships, which might contribute to future professional advantages (Cecić Erpič, Wylleman & Zupančič, 2004). Thus, compared to their counterparts enrolled in private universities, student-athletes attending public universities could show lower SM when prioritizing their academic careers to prepare for future job opportunities (Amara et al., 2004; McKenna & Dunstan-Lewis, 2004). Additionally, the Brazilian university sports facilitating sub-elite athletes’ participation in public institutions are still in a developmental phase (Starepravo et al., 2010).

At the international level, there is a call for awareness of the role and responsibilities of dual-career actors (Condello et al., 2019), and the implementation of support policies for student-athletes is a priority of the European Union (Aquilina & Henry, 2010; European Commission, 2012; EuropeanParliament, 2015; European Parliament, 2017). Indeed, student-athletes should be informed about their rights, and the implementation of dual-career counseling has been strongly recommended to help them manage their academic and sports commitment (Hansen & Sackett, 1993; Martin, 2005; López de Subijana, Barriopedro & Conde, 2015). During the academic path, a recognized student-athlete status could allow the monitoring of the academic and sport progresses to individualize necessary dual-career support. Although this status has been considered crucial to influence substantially the athlete’s motivation to achieve an academic degree, especially for those determined to pursue a sports career, the present models do not support this hypothesis. Thus, it is likely needed to take a step back and review the conceptions of support based on interactions at and across environmental levels (Knight, Hardwood & Sellars, 2018).

Study Four

To explore the SAMSAQ-PT scores’ sensitiveness, we considered repeated measures across an academic year was considered for the analysis of changes in motivation scores adjusted for sex, sport level, student-athlete status, and type of university. Given attrition expected in longitudinal observations, a Bayesian multilevel regression modeling and poststratification were used to predict the changes applied to all observations in the cross-sectional data. Bayesian multilevel regression and poststratification allow for improved estimations of small and sparse group data (in the present study, the longitudinal observations) and consequently predicts a target population (in the present study, the cross-sectional observations) (Gelman & Little, 1997; Park, Gelman & Bafumi, 2004; Kennedy & Gelman, 2020).

Methods

Participants

This study included 99 female and 35 male student-athletes enrolled in public (n = 68) and private (n = 66) Brazilian universities competing at international (n = 12), national (n = 39), state (n = 25), and university (n = 58) levels. Data were collected with a one-year interval (2018 and 2019) during the state and national University championships.

Data analysis

Due to the presence of non-representative and imbalanced data, with hierarchical sources of variation or cross-classified nesting, the first step of the analytical approach was to fit multilevel models to the repeated measures data, allowing for the possibility of varying intercepts (i.e., baseline values) and slope (changes in individuals outcomes across the period of observation) by sex, student-athlete status, sport level, and type of university (for individual i, with indexes, s, a, l, and u for, sex, student-athlete status, sport level, and type of university, respectively). Considering the homogeneity of slopes by group, we fitted varying intercepts models with measurement time as a population-level effect. The multilevel model specification was as follows: yi=β0sisex+αaistudent−athletestatus+αlisportlevel+αuitypeofuniversity+ϵi

αsisex∼N0,σsex2, for s = 1, 2.

αaistudent−athletestatus∼N0,σstudent−athletestatus2, for a = 1, 2.

αlisportlevel∼N0,σsportlevel2), for l = 1, 2, 3, 4.

αuitypeofuniversity∼N0,σtypeofuniversity2), for u = 1, 2.

ϵi∼N0,σyi2)

Then, the multilevel model estimates to predict the student-athletes’ outcomes for groups defined in a poststratification dataset (i.e., measurement time, sex, student-athlete status, sport level, and type of university) were used. The poststratification table considers the cross-sectional data sample in this research as a target population. The poststratification table has an observation corresponding to each group defined for all combinations of the variables included in the model. In this study, the poststratification table included two repeated measurement levels, two sex levels, two student-athlete status levels, four sport levels, and two type of university levels, encompassing 64 rows (2 × 2 × 2 ×4 ×2), including the sample size, in each group. After predicting the outcome variable for each group, estimates for measurement time with the subgroup sample sizes were aggregated. Hence, the method allowed full use of all available data to interpret the changes in student-athletes’ motivation scores, adjusted for individual and contextual characteristics. The model’s estimates were regularized by using normal prior (0,10) for population-level effect (intercept) and normal priors (0,1) for group-level effects. Four chains for 4,000 iterations with 1,000 burn-in iterations were run. The models were obtained using the “brms” package (Bürkner, 2017).

Results

Predicted changes for SM, AM, and CM, after an academic year and adjusted for sex, type of university, sport level, and student-athletes’ status are summarized in Fig. 4. The influence of the individual and context characteristics was similar at the baseline and after an academic year. The post-stratified predictions for the total sample showed a trend of stability for all student-athlete motivation scores after an academic year, although there was a slight decrease in SM.

Figure 4 Predicted changes for Sport Motivation, Academic Motivation and Career Motivation, after an academic year, adjusted for gender, university type, competitive level and student-athletes’ status (90% credible intervals).

Posterior values for Career Motivation dimension by sex, type of universities, competitive levels and student-athletes’ status (67% and 90% credible intervals).

Discussion

We explored the student-athletes’ motivation scores’ sensitiveness across an academic year adjusted for sex, sport level, student-athlete status, and type of university. The results showed that student-athletes’ motivation was stable across an academic year. In addition to limited variations observed across an academic year, uncertainty estimates were narrow after adjusting for sex, type of university, sport level, and student-athletes’ status. These findings suggest that SAMSAQ-PT may be sensitive to track individuals changes over time. Although sports burnout was beyond the aim of the present study, decreased SM over time could be considered an indicator in interpreting dual career paths (Sorkkila et al., 2018), monitoring university dual-career social contexts, or highlighting the needs of the dual-career implementation. The present predictive models indicate that the student-athletes’ motivation could be influenced by the university context and how it cares about the requirements necessary to combine sport and academic commitments, especially when the academic requirements increase towards graduation. Based on the present longitudinal observations, the university sports contexts appear not to motivate student-athletes in their dual-career. Future studies focusing on support dual-career policies could contribute to a sound interpretation of how institutions and managers deal with dual-career needs and services.

While in some countries, a well-systematized relationship between education and high-performance sport seems to determine the high motivation for an athletic career (Fortes et al., 2010; Simons, Van Rheenen & Covington, 1999), in European countries with distinct academic and sports systems, athletes are confronted with the choice of one of the two paths (European Commission, 2016). In Brazil, sports and educational systems are also distinct. In line with the development of university sports in Brazil, the awareness of the importance of supporting student-athletes dual-career increases (Guidotti, Cortis & Capranica, 2015; Stambulova & Wylleman, 2019). While the present results showed that student-athletes’ motivation does not vary substantially across an academic year, further longitudinal studies are needed to verify whether student-athletes are motivated towards a dual-career untill their graduation. Indeed, cooperation between the academic and sports sectors needs to be implemented for providing a supportive entourage for student-athletes.

General Discussion

Considering that contexts influence the motivation towards dual-career of athletes and its specificities in the development of guidelines to support student-athletes, and contribute to the discussion on student-athletes’ motivation (European Commission, 2012). The four experimental studies encompassing BEFA, BCFA, and Bayesian multilevel regression models in cross-sectional and longitudinal samples validated the psychometric structure and assessed the sensitiveness of the SAMSAQ-PT instrument in a Brazilian context. Furthermore, the present study presents methodological and practical implications. Indeed, cross-cultural variation and confidence in the measured outcomes need to be established when adapting a psychometric scale to a specific context.

Given the debate concerning the limitations and inappropriateness of null-hypothesis testing, “statistical significance” and using p-values (Amrhein & Greenland, 2018; McShane et al., 2019), the Bayesian statistics was deemed appropriate to provide a natural approach to account for different sources of inferential uncertainty (Kennedy & Gelman, 2020). Furthermore, in the present study, the analytical approach to validate a questionnaire using a multilevel modeling framework was considered appropriate to deal with common limitations of sports psychology research, such as noisy measurements, between-individuals heterogeneity, complex interactions between outcomes, and non-representative and imbalanced samples. The present data showed that the questionnaire scales were sensitive to the individual and contextual characteristics of the target population. Overall, the SAMSAQ-PT was a valid questionnaire for the Brazilian context and may be extended to Portuguese-speaking countries. Lastly, the study contributes to the propositions of career construction theories within university-student athletes (Rudolph et al., 2019). Finally, data, codes and supplementary tables from the present research are available at (https://osf.io/cpwdv/).

Regarding the practical implications, the present findings provide a view of the Brazilian educational system and how it might be related to the student-athletes’ motivation towards a dual career. In particular, relevant higher education stakeholders should cooperate in implementing regulations and policies fostering the development of dual-career for supporting student-athletes in combining sports and academic commitments. Considering that no variation was observed across an academic year, sports counseling or dual-career developmental programs should focus on first-year students. Likely highly motivated student-athletes since their first year of university will be engaged in a dual-career during higher education and beyond. Local and national sports policies should also consider the need to decrease sex-related differences by providing more opportunities for women in sports and increase media attention for public universities and women in sports.

Despite the advantages of a Bayesian multilevel modeling approach and a large sample of student-athletes, the present study presents some limitations. Only student-athletes competing in the University Sports Games were considered even though university students could compete in other Brazilian championships. Thus, further studies are needed to provide insights into the sport and academic development. Another limitation pertains to the lack of information on individual and contextual characteristics such as chronological age, academic course and year, or type of sport (team or individual) that prevented further in-depth analyses. Future research is envisaged to investigate the student-athlete background and transition from high school to higher education and their motivation to choose public or private universities related to their dual-career.

Conclusions

The present study assessed the validity of the Portuguese version of the SAMSAQ-IT/A for the Brazilian context and potentially can add to advance the understanding of the student-athletes’ motivation for a dual-career in other Portuguese-speaking countries. The use of the Bayesian estimations for psychometric analysis, multilevel regression and poststratification added to the analysis of the validity of psychometric scales in the study of student-athletes, which might suggest the revision of scales used in other countries. Based on the findings, it is possible to assume that individual and contextual characteristics need to be considered when investigating dual-career motivation. Based on the Brazilian context, the academic (public and private universities) and the sport (sport level) contexts may substantially impact student-athletes’ motivation. Although the student-athletes’ motivation overtime did not present substantial variation, the motivation decreased with time. Thus, the present results highlight the need to monitor the athlete’s motivation towards sports and academic achievements untill graduation to develop optimal dual-career paths. The final translated and validated version of the SAMSAQ-PT is presented in Appendix S1.

Supplemental Information

Supplemental Information 1 Questionnaire in portuguese

Portuguese version of the Student-athletes’ motivation toward sport and academics questionnaire

Click here for additional data file.

Additional Information and Declarations

Competing Interests

Author Contributions

Ethics

Data Availability

The authors declare there are no competing interests.

Ricardo T. Quinaud conceived and designed the experiments, performed the experiments, analyzed the data, prepared figures and/or tables, authored or reviewed drafts of the paper, and approved the final draft.

Carlos E. Gonçalves and Laura Capranica conceived and designed the experiments, authored or reviewed drafts of the paper, and approved the final draft.

Kauana Possamai and Cristiano Zarbato Morais performed the experiments, authored or reviewed drafts of the paper, and approved the final draft.

Humberto M. Carvalho conceived and designed the experiments, analyzed the data, prepared figures and/or tables, authored or reviewed drafts of the paper, and approved the final draft.

The following information was supplied relating to ethical approvals (i.e., approving body and any reference numbers):

The Federal University of Santa Catarina granted Ethical approval to carry out the study (CAAE: 96479318.8.0000.0121).

The following information was supplied regarding data availability:

The data and code are available at OSF: Quinaud, R, Gonçalves, CE, Possamai, K, Morais,CZ, Capranica, L and Carvalho, HM. 2021. “Validity and Usefulness of the Student-Athletes’ Motivation toward Sport and Academics Questionnaire: A Bayesian Multilevel Approach.” OSF. Data and Code. https://osf.io/cpwdv/.

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
