# Peer review of "Validity and usefulness of the student-athletes’ motivation toward sport and academics questionnaire: a Bayesian multilevel approach"

_PeerJ, doi:10.7717/peerj.11863_

## Round 0.1 · original submission · Major Revisions

Three consistent reviews have been received. Some merits are found by the reviewers. However, there are many issues that need to be clarified before it can be further considered. Please provide point-to-point responses.

·

Basic reporting

The manuscript presents the validation process of the SAM-SAQ/PT. The manuscript is well written and structured. The sources of information are updated and relevant.
The methodology and statistical analyses are its main strengths. The initial large sample allows the authors to conduct the validation, however, reusing with the same sample for further analyses maybe not necessary as the core of the contribution is the validation of the SAM-SAQ/PT tool.
Still, the content of this article would be a great contribution to the Dual Career topic, in case it is published.
I have made some specific suggestions which I hope the authors find them in a constructive way.
My recommendation for this manuscript is a major revision.

Experimental design

These comments are in the "specific comments".

Validity of the findings

These comments are in the "specific comments".

Additional comments

SPECIFIC COMMENTS
ABSTRACT
Review the content of the abstract based on the specific comments about the four studies.

L 25-26 & 28 Authors repeated the content " Four studies were performed."
L 31-32 I do not understand study 3
L 39 Predictions of scores ¿? I thought It was a comparison

INTRODUCTION
L 60 Authors introduce here for the first time the dual-career pathways, but this concept is not explained.
L 81-84 Please provide data about public-private universities. Which is the proportion of them in the whole country? and what is the number of students in each one?
L 84 & L 88 These references are not included in the final list. Please provide them.
L 92, 95 & 96 Please revise the references.
(Open Science Collaboration 2015) needs a comma.
(Amrhein & Greenland 2018, needs a comma
McShane, Gal et al. 2019) should be McShane et al., 2019)
AIMS
L114-119 Please writ-state each aim (one per study) separately in this paragraph rather than "addressing".
L116-117 the SAM-SAQ IT is supported by the reference (Guidotti & Capranica, 2013; but the "A" is not included in that reference. Please revise the whole manuscript.

METHODS
I would recommend authors changing this section structure.
I had problems understanding the section itself.
First, the information is repeated or the content is not in an adequate place. For example:
• L 131-133 there are some variables mentioned that should be included in the tools subsection.
The research design could include the ethics committee information and the inclusion criteria of the different samples of the study.
Then in each study authors could follow the structure of:
Participants- Tools-Procedure-Data analysis- Results-Discussion

L 156-157 No supplemental files are available for review.
L 157- could you give more information about the sample? How many student-athletes compete in those championships? What was the response rate? Were they representative of their population? Were they from all the regions of Brazil? What was their race? What kind of studies they were enrolled in? STEM (Sciences Technology Engineer and Mathematics), Social Sciences, Humanities, Life Sciences…

DISCUSSION study 1
I would suggest authors begin each discussion with the aim of the study and the main results found.
L 180-189 Authors should focus on the aim of this study. These lines explain each factor which could be explained in the introduction section. Authors should compare their results (3-factor solution) with the Italian, European, and Asian validation processes.
L 189-190 This speculation should be based on the results. I cannot find the link with the Exploratory Factor Analysis results.

STUDY 2
Line 222 Please explain from the 17 final items which are from SM, AM, and CM.
L 225 Supplementary Table 2 is not available to revise.
L 237-238. Could you please detail the main differences found between this SAMSAQ-PT and the other languages versions?

STUDY 3
The aim of this study is to discriminate (separe) or to compare by gender, type of university, sport level…?
Later in the paper at the study 3 results (L 282) authors state the differences found by gender….
L 243-244- Please write the independent variables as gender, sport level, type of university and athlete status.
Please explain clear the athlete status meaning. In the tools section not in data analysis.
L 247-249 In study 1 and 2 authors validated the SAMSAQ/PT with this sample. The reuse of the sample for other purposes is not justified.
L 281-284 Please divide the differences found for each factor. With this writing, it is difficult to understand the message.
For example in Figure 1 at Sport Motivation female had higher z-score levels than male, while in the text they state just the opposite. Please revise each factor and the differences found.
Another option and based on the discussion structure would be to divide this text into gender differences- sport level differences-type of university differences and student-athlete status differences. But clearly in the text, please do not mix both writings (factors with independent variables).
L 284 Supplementary Table 3 is not available to verify its content.
Why using bayesian models when you only want to compare by one independent variable?
L 289 please begin each discussion with the aim of the study and the main results.
Structure this discussion following the same structure as the results.
L 292- 294 Please revise the English grammar ("their sport one"¿?)
L 295 Please omit writing identity when measuring motivation.
Please revise the last name of these references:
L 319- Cecic Erpic and L 331 López de Subijana

STUDY FOUR
L 340- I do not agree with the authors when they propose a repeated measured design when they measured twice. Repeated measures is a concept applied to more than 2 measures.
L 351-354- I have the same concerns about re-using the sample for validating and for other studies in the same article.
L 357- Please check the writing (mon-representative).
L 386-387- I do not see these results in the Figure. They are supporting later the discussion and they are not in the Figure.
L 387 please remark clearly that the individual is gender while academic context is the type of university. They seem new variables in the text.
Discussion
Please structure the discussion as mentioned before. First, relate the results with the aim of the study.

L 396-397 Please limit the statement to the results of this study. The decrease of SM was not reported in the results section.
L 399-402 I could not appreciate this result in Figure 4 of this study. Please limit the content to the results found.

GENERAL CONCLUSIONS

L 349-340. This sentence when this theoretical framework (Rudolph et al., 2019) is not included in the introduction does not increase
the manuscript contribution.
L 440-441- I suggest authors include the data repository in….
L 444-446 This recommendation is for fostering a dual career in public universities?
L 457- I do not understand why the authors acknowledge chronological age as a limitation when it was informed in line 154.

I think authors should limit their manuscript to the main contribution: validating the SAM-SAMQ/PT.
Reusing the sample for other purposes I think is not appropriate.

Reviewer 2 ·

Basic reporting

The authors present a manuscript of relevant interest. The subject falls within the scope of the journal. However, the text still needs some amendments, clarifications, and corrections.

Experimental design

Are there really four studies, or is it a single study with four phases?

Validity of the findings

no comment

Additional comments

(1) Avoid unnecessary repetition of words and phrases. Examples:
(1.1) In the abstract: "The purpose of this research was to examine the validity and usefulness of the student-athletes`motivation toward sport and academics questionnaire (SAMSAQ-PT) in Brazilian student athletes".
(1.2) In the abstract: "Four studies were performed" is written twice.

(2) Are there really four studies, or is it a single study with four phases?

(3) Lines 98 and 431, change dierent to different.

(4) Line 154: "age 21.7 ± 3.4 yrs". The notation “mean (SD)” should be used throughout instead of “mean plus/minus SD,” since this is not a confidence interval. In other words, the SD should be indicated in parentheses.

(5) Line 163. Is a total of 20000 samples sufficient for the convergence of the MCMC algorithm? How is this verified?

(6) Line 164. What is default identification restriction? A brief explanation of its meaning and usefulness should be inserted into the article.

(7) Line 165: "The minimum posterior means for a factor loading to be considered as a benchmark was set as 3". Is this a usual choice? Are there bibliographic references for this?

(8) Line 171: "Due to low factor loadings, items 11 (2.70), 18 (2.09), 25 (2.44), and 30 (1.78) were excluded". What value determines a low factor loading?

(9) Line 206: "Two chains for 10,000 iterations with 2,000 burn-in iterations and the default Stan were run". What is Stan?

(10) Line 207: "The model estimations were regularized using normal prior (0, 10) for the manifest variable (intercept) and normal prior (0, 1) for the latent variable were set". What motivated the choice of these hyperparameters? Are they non-informative priori distributions? Are these recommendations from the literature?

(11) Line 253: "...Brazilian student-athletes when grouped by gender (e.g., female and male)...". Sex and gender are different concepts, sex refers to biological characteristics and gender refers to the individual's and society's perceptions of sexuality. Do the authors believe that this differential is important in the context of this paper?

(12) Lines 269 and 380: "Two chains for 4,000 iterations with a warm-up length of 1,000 iterations to ensure convergence of the Markov chains were run". Are the authors sure that 4000 MCMC samples are sufficient for chain convergence? It seems very little to me. As a rule of thumb, we can observe if the MC error is less than 1% to 5% of the corresponding posterior standard deviation.

(13) What is the motivation for using 67% and 90% credible intervals?

(14) Line 269: "Two chains for 4,000 iterations...". Line 379: "Four chains for 4,000 iterations...". Why did one Bayesian model use 2 chains and the other used 4 chains?

(15) Was the Gelman-Rubin convergence diagnostic used?

Reviewer 3 ·

Basic reporting

Comment on language and grammar Issues.
• Line 98 “di*erent” “different”
• Line 131 “analyses” “analysis”
• Line 153 “dataset” “data set”
• Line 357 “mon-representative” “non-representative”
• Line431 “di*erent” “different”

Experimental design

• Consider an even number (0.2) or (2) as the variant rather than (.05) which appears to affect some of the accuracy in each of the four models.

• The authors Discussion section provide accurate next steps for further research.

Validity of the findings

No criticism. Great job on a difficult topic to quantify.

Additional comments

• Good job of introducing the questionnaire in the initial 4 lines of the background section.
• The methods are clearly defined
• Results are clearly defined
• Dual Career is clearly defined in lines 51,52,53, 54
• Categories pertaining to dual career are specified: Motivation, identity, health, lifestyle, and well-being.
• The historical acknowledgement of the SAMSAQ-PT provides a front loading to the information being introduced and analyzed.
• Literature references are sufficient and provide support for the research. Both the findings and what needs are necessary for improvements.
• Figures and analysis table are clearly explained and comprehensible
• Provides evidence of Portugese’s version of the SAMSAQ-IT/A
• Demonstrates the significance and importance of the research into increased support for Dual Career athletes.
• Great mastery of writing in the English language. Masterful use of writing conventions

---

## Round 0.2 · Minor Revisions

The paper has been improved but there are still some changes needed. Please revise the paper according to the reviewers' suggestions.

·

Basic reporting

no comment

Experimental design

no comment

Validity of the findings

no comment

Additional comments

Dear authors,
I find the manuscript better structured now. I would like to thank them for the effort done at this reviewing round.

- Still there are some minor issues in relation to typing errors at some references. Please check the following references in lines: 99, 102, 103, 110, 113, 118, 346, 347, 385,
- L 161- About the supplementary tables I would recommend authors to include this link each time the refer to supplementary materials as It won’t be available each content in the article.
- L 536-537 Please mention again that the supplementary tables are in that link.

Reviewer 3 ·

Basic reporting

This research utilized a Psychometric Structure of the SAMSAQ-IT/A (SAMSAQ-PT).

Four latent factors were ultimately used in four variant phases of the Research method and participants used or each.

Cognitive interviews were performed.

Professional language is utilized.

The data charts were clear and congruent with the written description.

Structure conforms to peer J standards

Experimental design

Cross sectional and longitudinal samples. Bayesian method combining information previously known and shared amongst all participants. The Portuguese version of (SAMSAQ-IT; Guidotti &Capranica,2013).

The four phases were detailed and results were explained.

This is within the scope of Peer J Journal. Investigation demonstrated time and effort.


The three primary categories for the student-athlete responses were:
Academic motivation, Sports motivation, and Career motivation. Likert scale was implemented also.

Rigorous analysis in connection with the overall scope of the research is evident.

Validity of the findings

Information is organized.

Number of participants was appropriate for the research. Not all participants were involved in each of the sections. This may be due to the varying levels of athletic participation by each athlete. Some in championship season and others (public university, private school) having different times of availability.

Simple background questions may be considered to create a more in-depth visual of the plight of the student athlete.

Additional comments

The research presented does contribute to career construction theories.

Comprehensive issue. Writers show knowledge of statistics and are able to translate into meaningful language.

The purpose of the research is justified.

Introduction paragraphs three and four present validations for the research and its use of the SAMSAQ.

Data charts are correctly labeled.


Suggested Edits
Line 52 foci- focus
Line 102 di^erent -different
Line 473 till- until
Line 498 di*erent- different
Line 532 till- until
Line 742 modelling-modeling

Reference section: has several article names that are unusually spaced out. Refer to: Lines 737, 740, 755, 766, 770, 782, 785 and 764.

Recommendations from research/Validity of Findings:

1. Sports counseling or dual career programs should focus on first year student-athletes
2. Provide more opportunities for women involved with sports
3. Samples were robust, statistically sound and, & controlled.
4. Create more media attention for public universities and women sports

---

## Round 0.3 · accepted · Accept

I have checked the revised paper. All required changes have been appropriately dealt with. Therefore, I am happy to recommend it for publication.